

# *Aspergillus niger* as an efficient biological agent for separator sludge remediation: two-level factorial design for optimal fermentation

Paveethra Thegarathah[1], Jegalakshimi Jewaratnam[1], Khanom Simarani[2] and Amal A.M. Elgharbawy[3,4]

[1] Department of Chemical Engineering, Universiti Malaya, Kuala Lumpur, Malaysia
[2] Division of Microbiology, Institute of Biological Sciences, Universiti Malaya, Kuala Lumpur, Malaysia
[3] International Institute for Halal Research and Training (INHART), International Islamic University Malaysia, Kuala Lumpur, Malaysia
[4] Department of Chemical Engineering and Sustainability, International Islamic University Malaysia, Gombak, Kuala Lumpur, Malaysia

Corresponding author
Jegalakshimi Jewaratnam,
jegalaxmi24@um.edu.my

## ABSTRACT

**Background**. The booming palm oil industry is in line with the growing population worldwide and surge in demand. This leads to a massive generation of palm oil mill effluent (POME). POME is composed of sterilizer condensate (SC), separator sludge (SS), and hydro-cyclone wastewater (HCW). Comparatively, SS exhibits the highest organic content, resulting in various environmental impacts. However, past studies mainly focused on treating the final effluent. Therefore, this pioneering research investigated the optimization of pollutant removal in SS *via* different aspects of bioremediation, including experimental conditions, treatment efficiencies, mechanisms, and degradation pathways.

**Methods**. A two-level factorial design was employed to optimize the removal of chemical oxygen demand (COD) and turbidity using *Aspergillus niger*. Bioremediation of SS was performed through submerged fermentation (SmF) under several independent variables, including temperature (20–40 °C), agitation speed (100–200 RPM), fermentation duration (72–240 h), and initial sample concentration (20–100%). The characteristics of the treated SS were then compared to that of raw sludge.

**Results**. Optimal COD and turbidity removal were achieved at 37 °C 100 RPM, 156 h, and 100% sludge. The analysis of variance (ANOVA) revealed a significant effect of selective individual and interacting variables ($p < 0.05$). The highest COD and turbidity removal were 97.43% and 95.11%, respectively, with less than 5% error from the predicted values. Remarkably, the selected optimized conditions also reduced other polluting attributes, namely, biological oxygen demand (BOD), oil and grease (OG), color, and carbon content. In short, this study demonstrated the effectiveness of *A. niger* in treating SS through the application of a two-level factorial design.

## INTRODUCTION

The global expansion of the oil palm agribusiness has resulted in the massive generation of liquid waste known as palm oil mill effluent (POME). Most palm oil mills add water during the extraction of crude palm oil (CPO) from fresh fruit bunch (FFB), which generates substantial amounts of POME and is stored in open ponds. An estimated 2.5–3.8 tons of POME waste may be produced from each ton of CPO during industrial processing, of which the majority is released into the environment without adequate treatment. POME is viscous and thick, produces an unpleasant smell, and has high colloidal suspension (*Syahin, Ghani & Loh, 2020*). As such, it is regarded as one of the most harmful pollutants that can negatively impact the ecology (*Ratnasari et al., 2022*).

Generally, POME is made up of sterilizer condensate (SC), separator sludge (SS), and hydro-cyclone wastewater (HCW) with a proportional ratio of 9:15:1 (*Alam et al., 2022*; *Wu et al., 2010*). A recent review article presented the characteristics of each component and highlighted the properties of SS, which possesses the highest amount of biological oxygen demand (BOD), chemical oxygen demand (COD), total suspended solids (TSS), total nitrogen (TN), ammoniacal nitrogen (AN), and oil and grease (OG) (*Wu et al., 2010*). Digested CPO is pumped into a clarity tank to separate the oil from the CPO at 90 °C to enrich the oil separation process further. The composition of the digested CPO is estimated to be 35–45% palm oil, 45–55% water, and residual fibrous debris. Subsequently, the oil at the top of the tank is skimmed off. Meanwhile, the bottom phase of the clarity tank, which includes residual oil, goes through the decanter heavy phase (DHP) tank and the de-oiling tank of the SS once again. This purification process produces sludge, which is a major source of POME in terms of the quantity and degree of pollution (*Mohammad et al., 2021*).

The need for large disposal sites, unpleasant odors, energy-intensive, and high operating costs are among the several examples of sludge-related issues (*Ahmad Farid et al., 2019*; *Jemaat & Qi, 2022*). While research efforts are ongoing to treat POME, most of the bioremediation studies focused on POME instead of the most hazardous component in POME, particularly SS. Thus, a greater emphasis should be given to treating SS, given that its high organic content could affect flora, fauna, and the environment in various ways (*Chaturvedi et al., 2021*; *Parida et al., 2021*; *Sathishkumar et al., 2020*). According to past literature, the integrated anaerobic-aerobic bioreactor (IAAB) system has been found as one of the most efficient modern alternative treatment methods, as it reduces COD and BOD within 115 days (*Mohammad et al., 2021*). Nevertheless, the treatment duration is disproportionate to the waste accumulation. Moreover, operational difficulties could arise from effluent overflow, system backups, or the requirement for extra storage sites if the treatment process takes longer than the accumulation rate.

The most common bioremediation occurs through biodegradation. Microorganisms, such as bacteria and fungi, can break down complex organic pollutants into simpler, less harmful compounds. As an economical and environmentally beneficial approach to reducing contaminants, bioremediation offers many advantages. For example, bioremediation provides a focused solution to treat various contaminants, minimizes

long-term responsibilities, uses less energy, and produces less secondary waste when operating in different environmental circumstances. On top of that, bioremediation decontaminates the target substances through microbial metabolism, producing enzymes and metabolites without harming the environment (*Jabbar et al., 2022*). Since fungi can tolerate higher amounts of contaminants than algae, archaea, and bacteria, they have an advantage over other biological remediation agents (*Ali et al., 2017*; *Kumar et al., 2021*; *Rezania et al., 2016*).

One of the most prevalent fungal contaminants in vegetables is *Aspergillus niger*, which also has numerous industrial uses, including in the production of citric acid (CitA), cellulases, lipases, amylases, proteases, and wastewater treatment. This is because *A. niger* can break down agricultural waste in an environmentally beneficial manner. As an asexual saprophyte, *A. niger* can grow in almost any aerobic habitat. Aside from its resilience to heat, which allows it to grow in freezing and extremely hot conditions, *A. niger* can reproduce and multiply within a temperature range of 6–47 °C and grows best at pH 6, although it can tolerate a wide pH range from 1.5 to 9.8. Furthermore, the growth of this species is optimal at a relative humidity and water activity of 0.97 and 96–98%, respectively (*Lima et al., 2019*). Given that *A. niger* is one of the indigenous fungal species found in POME (*Bala et al., 2018*), *A. niger* could be utilized as a bioremediation agent in degrading SS through submerged fermentation (SmF), which enables easier scale-up, product recovery, and better monitoring and control of the treatment process.

Considering the fact that bioremediation is influenced by both abiotic and biotic elements (*Zhang et al., 2020*), it is crucial to identify the range of factors, such as the fermentation duration, shaking speed, temperature, and raw sample concentration, to ensure optimized remediation by *A. niger*. Based on these premises, this study was conducted to assess the impact of bioremediation of *A. niger* on the physicochemical properties of SS in POME and quantify the key components and their interactions with a minimal experimental run using a two-level factorial design. The proposed simple and rapid SmF technique to treat SS in this study is based on the presence of high organic and solid content in SS. The results obtained would suggest optimum sludge treatment under specific conditions. Besides, characterization of the obtained residue should provide basic information about the metabolites present as evidence of revalorization during degradation. As the ultimate goal of bioremediation is to convert sludge into biomass or non-toxic metabolic products, at the very least, this study is considered a pioneer revelation of bioremediation and revalorization of SS from POME for sustainable and economical sludge handling in the future.

## MATERIAL AND METHODS

### Sample collection

The SS sample was collected from a palm oil mill facility owned by Sime Darby Research Sdn. Bhd. in Carey Island, Selangor, Malaysia and serves as the raw material in this study. The sample was stored in sterile containers and refrigerated at 4 °C to avoid contamination and deterioration. Subsequently, the sample was analyzed in terms of COD, turbidity, pH,
BOD, TSS, OG, color, total carbon (TC), total organic carbon (TOC), and total inorganic carbon (TIC).

## Inoculum preparation

This research used black Aspergillus (*A. niger*) stock culture from the Microbiology Department, Institute of Biological Science, Faculty of Science, University of Malaya. The fungal strain was revived by sub-culturing onto fresh potato dextrose agar (PDA) for microbiology, EMSURE® ACS, ISO plates. After the culture plates were incubated at $30 \pm 2\,°C$ for 7 days, the culture was scraped with a sterilized spatula, and the spores were reconstituted in sterilized distilled water. The spore mixture was homogenized by shaking using a vortex mixer before filtering the spore suspension *via* cotton gauze to remove hyphal filaments. The spore suspension stock with $1 \times 10^5$ cells/mL was then prepared and maintained at $4\,°C$ for further use. Note that the spore count was measured using a Neubauer's chamber (hemocytometer). Approximately $100\,\mu L$ of the spore suspension was loaded into the hemocytometer chamber. No dyeing material was required to stain the spores due to their black color nature. Spore suspension were counted in triplicate.

## Design of experiment

Table 1 shows the four fermentation factors examined in this study, including temperature, agitation rate, fermentation duration, and sample concentration. Each parameter was chosen because they have significant effects on fungal development, play a crucial part in the bioremediation process, and can be optimized while ensuring a cost-effective outcome (Table 2). For instance, the temperature was examined in the range of 20–40 °C, which corresponds to the ideal growth conditions for fungi and to avoid the inhibitory effects at extremely high or low temperatures. The selected agitation range of 100–200 RPM was applied as the best possible aeration, which prevents the negative consequences of over-agitation. In addition, the SS concentration range of 20–100% was chosen to balance the requirement for nutrients with the effects of water availability on fungal activity. Finally, the fermentation duration of 72–240 h was selected with the intention of optimizing bioremediation under a short duration, considering the unique characteristics of the SmF process and microbial activity. It is noteworthy that the pH level and nutrient addition were excluded from the optimization process since pH modification requires adding other chemical constituents, which does not align with cost-effectiveness, and adequate nutrients are present in the selected sludge.

A two-level factorial design is a structured and efficient approach in experimental design and statistical analysis to investigate the effect of multiple factors on a particular response variable. In this method, the factors of interest are manipulated at two levels, typically high and low, to explore their impact systematically. The main effects (the individual influence of factors) and interactions (how these factors influence each other's effects) are identified by evaluating all possible combinations of the factor settings. This approach significantly reduces the number of experiments required to optimize the process, product, or system, making it a valuable tool for decision-making and quality improvement (*Anderson & Whitcomb, 2015*).
**Table 1  Parameters for the two-level factorial design.**

| Factor | Name | Units | Type | Minimum | Maximum |
|---|---|---|---|---|---|
| A | Temperature | °C | Numeric | 20.00 | 40.00 |
| B | Agitation speed | RPM | Numeric | 100.00 | 200.00 |
| C | Fermentation duration | Hours | Numeric | 72.00 | 240.00 |
| D | Sample concentration | % | Numeric | 20.00 | 100.00 |

**Table 2  Independent variables: selection criteria and justification.**

| Parameters | Selection criteria | Justification |
|---|---|---|
| Temperature | Ideal temperature range for the fermentation process and fungal development. | Temperature has significant effects on metabolic and microbiological activity. It influences growth rates as well as on compound transport and saturation levels inside cells. |
| Shaking speed | Impact on shear forces on fungal cells, dissolved oxygen, and mixing efficiency. | The culture medium's mixing and oxygen availability are influenced by the shaking speed. Higher speeds may result in shear forces that harm cells, while lower speeds may create oxygen deprivation that prevents growth. |
| Sample concentration | Impacts on yield, rate of fermentation, and stressors associated with substrate supply. | Product yield and fermentation efficiency are impacted by substrate concentration. While higher concentrations can initially result in better rates, they can also cause inhibition beyond an ideal range. |
| Fermentation duration | The ideal period to reach the targeted bioremediation efficiency in a feasible time duration. | The duration of fermentation plays a pivotal role in achieving the desired yield and quality in the bioremediation process. While longer fermentation periods initially facilitate increased microbial activity for contaminant degradation, there's a point of diminishing returns. Extended durations can lead to decreased degradation rates, plateauing microbial activity, and the development of resistant compounds, limiting further breakdown. Balancing the fermentation duration is crucial to optimize remediation efficiency without unnecessary resource utilization. |

**Notes.**

These selection criteria and their respective justifications table provide a clear rationale for choosing these independent variables in the study. This study has considered the impact of environmental factors on fungal growth, fermentation efficiency, and bioremediation goals while aiming for a cost-effective and optimized process.

For this study, a two-level factorial design consisting of 16 experimental trials ($2^4 = 16$) with duplicates was conducted in a randomized manner to evaluate the four variables at two coded levels. The COD removal (%) and turbidity removal (%) were recorded as the two response variables. The results were presented as the mean and standard deviation (SD). Microsoft Excel® for Microsoft 365 and Design Expert 11.0 software® (version 11.1.2.0, Stat-Ease Inc., Minneapolis, MN, USA) were employed to perform statistical analysis and process the results. The statistical significance of the model was ascertained using the analysis of variance (ANOVA).

## Submerged fermentation and operational conditions

The fermentation process was conducted in 250 mL conical flasks containing 100 mL of the fermentation media (SS) at various concentration ranges of 20–100%. Each medium concentration was prepared with distilled water, and the media were sterilized at 121 °C

(15 psi) for 15 min. Once the media were cooled, each flask was inoculated with one mL of the *A. niger* spore suspension and incubated on a rotary shaker at a specific speed and temperature (100–200 RPM and 20–40 °C) for several durations (72–240 h). Each condition was performed twice. The fermentation slurry was collected and centrifuged at 10,000 RPM for 15 min at 4 °C.

## Analytical method

The COD and the turbidity of the fermentation slurry were assessed using a portable colorimeter (DR/890). In addition, a tabletop pH meter (Mi 151) was used to determine the pH of the sample and the fermentation slurry. The five-day BOD ($BOD_5$) method was applied to measure the BOD using the protocol specified in section 4500-O.G. of the American Public Health Association (APHA) 5210 (*APHA, 2005*). Briefly, the sample was filtered with Whatman#1 filter paper, rinsed twice with deionized water, and dried to a constant mass at 105 °C to collect the TSS. The dry solids' mass was then calculated. Besides, the OG was calculated using the standard technique outlined in APHA 5520 G. The sample's color was measured using a spectrometer from Avantes. The carbon content (TC, TOC, and TIC) was assessed using Shimadzu's TOC-LCPN, TNM-L, and SSM-5000 total organic carbon analyzer. Eq. (1) (*Tak et al., 2015*) was used to calculate the removal efficiency (*R*):

$$R(\%) = \left[ 1 - \frac{Y}{Y0} \right] * 100 \tag{1}$$

where $Y_0$ and $Y$ represent the initial (raw sludge) and final values (treated SS supernatant) of COD, turbidity, BOD, OG, color, TC, TOC, and TIC, respectively.

## Verification of the model

Further experimental work was done under optimized experimental settings as recommended by the response surface methodology (RSM) to validate the expected outcomes from SmF. To determine the adequacy of the models, the acquired experimental values were compared to the predicted values given by the models. Then, the predicted error analysis was conducted using Eq. (2):

$$\text{Predicted error } (\%) = \left( \frac{\text{Experimental value} - \text{Predicted value}}{\text{Predicted value}} \right) * 100. \tag{2}$$

## Characterization of the raw and treated SS

The surface morphology and chemical functional groups of the raw and treated SS were determined using a Scanning Electron Microscope (SEM) and Fourier Transform Infra-red (FTIR) spectrometer-spectrum 400 (Perkin Elmer, Waltham, MA, USA).

# RESULTS AND DISCUSSION

## Half-normal plots, pareto charts for standardized effect and analysis of variance

A two-level factorial design was employed in this study to identify the variables with the most significant influence on the removal of turbidity and COD. Half-normal plots were

used to assess the robustness of the model selection. By ranking the absolute values of the various effects, the half-normal plots for COD and turbidity removal present the critical components influencing the responses, as shown in Fig. 1. All factors below the regression line (red) have negligible effects on the responses. In contrast, the other factors and their interactions with one another have a significant impact. Higher relevance to the responses is shown by further separation from the line (*Tuminoh, Hermawan & Ramlee, 2021*).

The half-normal plots in Fig. 1 also demonstrate that nearly all the primary factors (A (temperature), B (agitation speed), C (fermentation duration), and D (sample concentration)) and interacting components are situated to the right of the normal line, highlighting the significance of degradation. Intriguingly, factors A and D are far to the right in Figs. 1A and 1B, respectively, indicating the crucial impact of temperature in reducing COD and the sample concentration in removing turbidity. On the other hand, the prevalent interaction elements on the right show their favorable effects on degrading efficiency. Additionally, all factor bars (Fig. 2) exceeded the *t*-value limit (2.11991) for COD removal, demonstrating their significance. However, the bars of the main component C and the interacting factors ACD, BD, and AC in the Pareto chart obtained for turbidity removal are below the *t*-value limit. The method was also used to determine the effects' values, which were compared to the reference lines' typical *t*-value and the stricter Bonferroni limit (3.44432). According to *Myers, Douglas & Anderson-Cook (2016)*, all effects above the Bonferroni limit are significant-to-crucial, while those above the t-limit may be significant-to-moderately important. Thus, the individual factor D and the interacting terms AB, BC, CD, ABD, and ABCD significantly influenced both responses.

In terms of COD removal, the interactive factor ABC was the main causing factor with 23.44%, followed by ACD (21.18%) and BD (16.82%) (Table 3). On the contrary, the interactive factor ABCD recorded the highest turbidity removal of 22.71%. Table 4 displays the ANOVA findings on the selected regression model from the two-level factorial design to identify the significant main and interaction effects of the factors impacting deterioration. Fisher's F-test, which refers to the distribution of the ratio between the respective mean square effect and mean square error, was utilized to determine the significant difference in relation to the control response and to compute the standard errors. The experimental parameters that were statistically significant on a given response were determined using the *p*-values. The analysis was conducted at a 95% confidence level for model prediction, indicating that variables with a *p*-value of 0.05 or less significantly affect the response.

Based on the *p*-values obtained by ANOVA, all the factors were statistically highly significant for COD removal except for factors C and D, which are 0.3118 and 0.3021, respectively. Despite the significance of the turbidity removal model, many main and interactive factors, including factors B, C, AC, BD, and ACD, were insignificantly different, with *p*-values higher than 0.05. Furthermore, the F-values for Responses 1 and 2 also showed a high significance of 729.65 and 14.01, respectively. An F-value with a low probability of occurrence ($p < 0.0001$) indicates that there is a 0.01% possibility caused by noise. It is evident that the first-order polynomial equation was statistically significant, which could be used to predict the removal of COD and turbidity from SS.

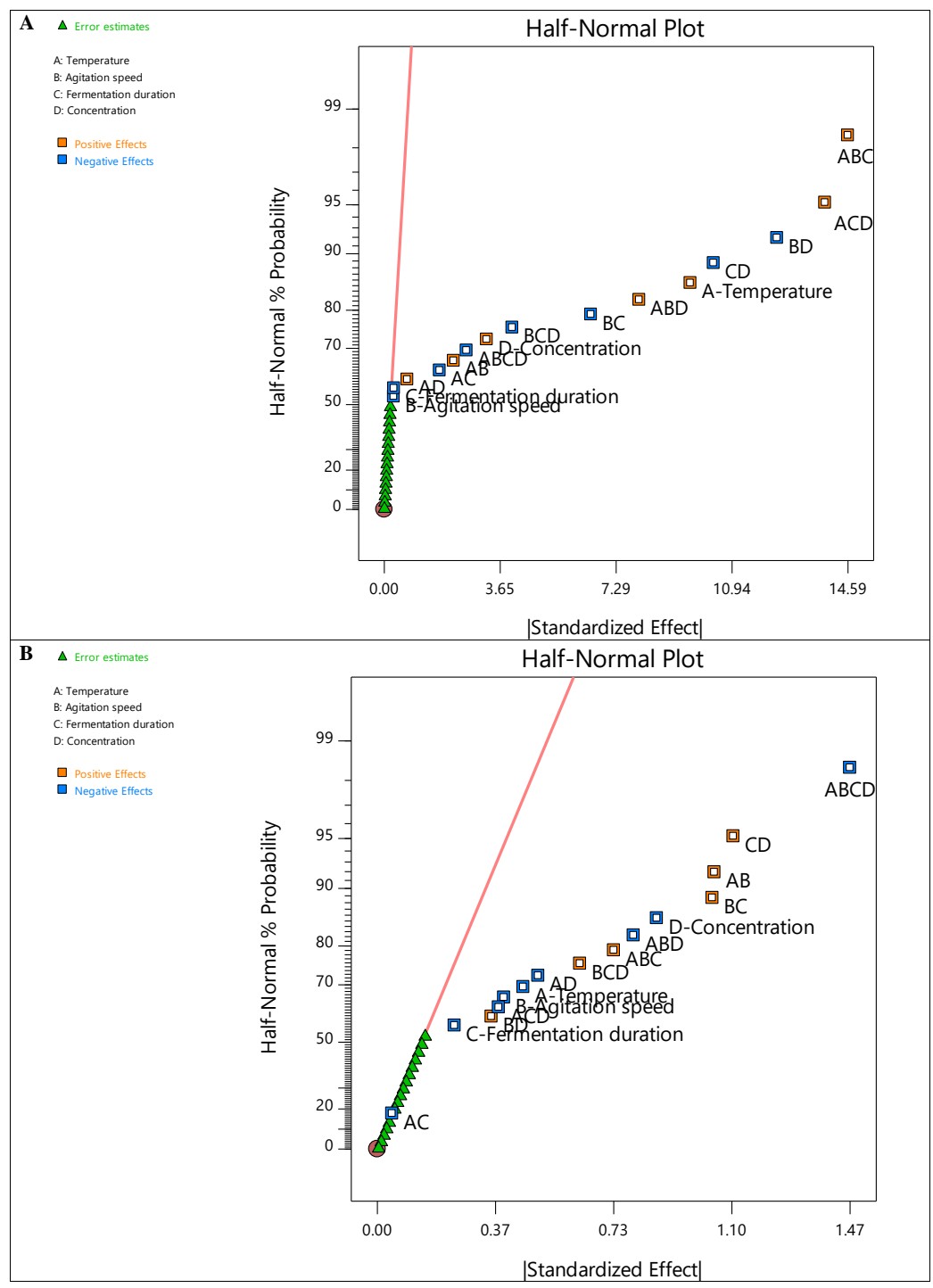

**Figure 1 Half normal plot graph for (A) COD removal and (B) turbidity removal.**

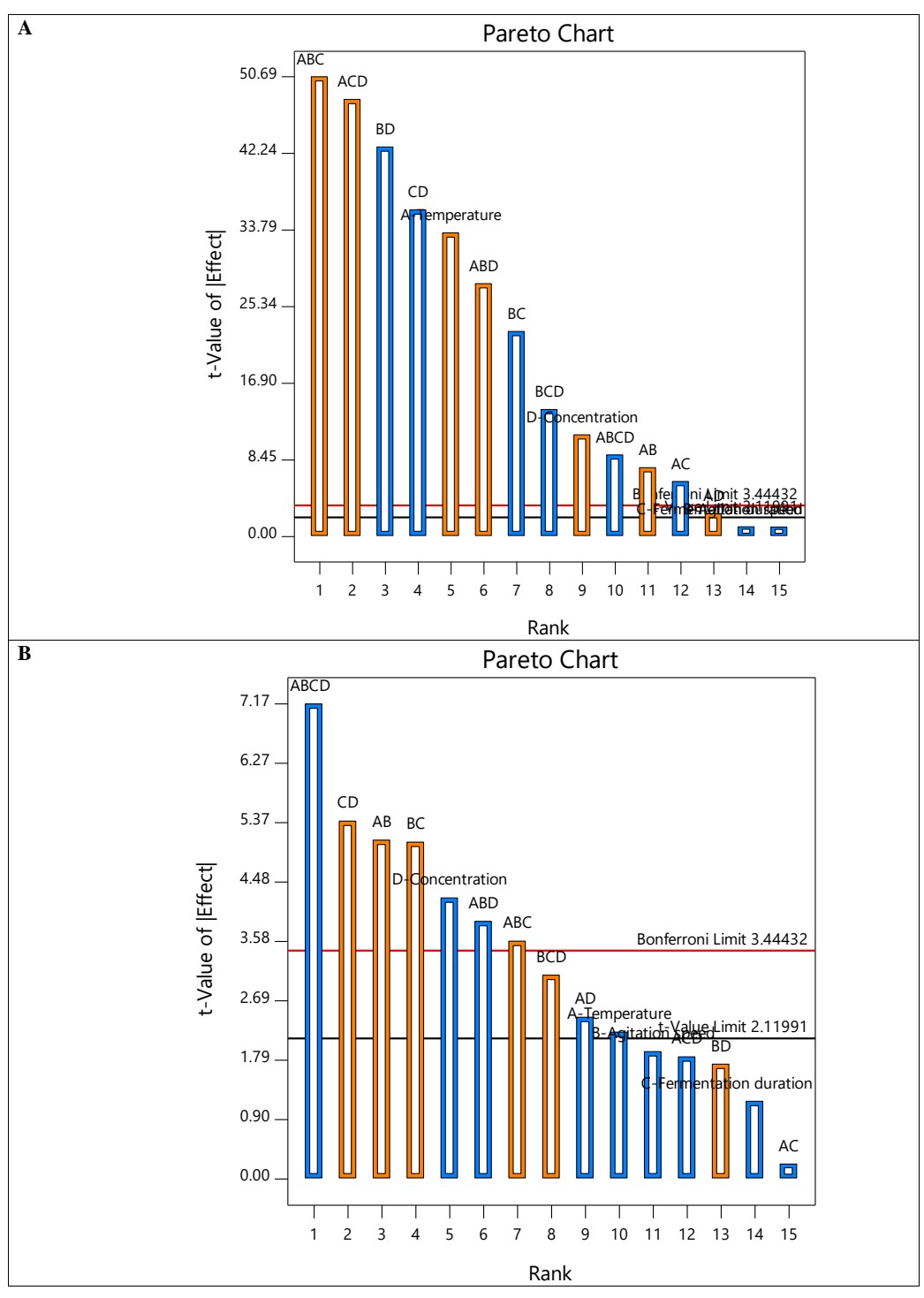

**Figure 2** The Pareto chart of all factors and interactions on (A) COD removal and (B) turbidity removal.

**Table 3  Percentage contribution of factors on responses.**

| Factors | Percentage contribution (%) | |
|---|---|---|
| | Response 1 | Response 2 |
| A | 10.2178 | 2.16729 |
| B | 0.00995374 | 1.63202 |
| C | 0.0103719 | 0.606327 |
| D | 1.14683 | 7.94464 |
| AB | 0.526123 | 11.5503 |
| AC | 0.335612 | 0.0225169 |
| AD | 0.057991 | 2.63163 |
| BC | 4.66138 | 11.4129 |
| BD | 16.824 | 1.3266 |
| CD | 11.8282 | 12.8822 |
| ABC | 23.4388 | 5.68668 |
| ABD | 7.08742 | 6.67396 |
| ACD | 21.1822 | 1.50009 |
| BCD | 1.78818 | 4.17797 |
| ABCD | 0.739172 | 22.7081 |

**Table 4  ANOVA for COD and turbidity removal.**

| Source | Sum of squares | df | Mean square | F-value | p-value | |
|---|---|---|---|---|---|---|
| Model-COD removal | 7253.03 | 15 | 483.54 | 729.65 | <0.0001 | Significant |
| Model-Turbidity removal | 70.62 | 15 | 4.71 | 14.01 | <0.0001 | Significant |

**Notes.**
df, Degree of freedom.

## Fit statistic calculation

The coefficient of determination ($R^2$) and adjusted $R^2$ value for the COD removal were 0.9985 and 0.9972, respectively. The values in Table 5 demonstrate a strong correlation between the experimental results and the model predictions. The nearly identical $R^2$ and adjusted $R^2$ values indicate that the model does not contain negligible variables (*Nacer et al., 2021*), elucidating the significance of the studied factors (temperature, agitation speed, fermentation duration, and sample concentration). In other words, adding significant independent variables to the model will increase the adjusted $R^2$ value, while adding non-significant ones will reduce the adjusted $R^2$ value. Hence, including more variables offers no improvement in the COD removal. Conversely, the adjusted $R^2$ value will increase regardless of the significance of the additional variable. The $R^2$ value also indicates that the first-order polynomial regression model under consideration could account for 99.85% of all changes in COD removal. Furthermore, the predicted $R^2$ of 99.42% shows that the model could measure the predicted response accurately. Using the standard error of the mean (SEM), the value was applied to calculate the standard deviation of residuals, which was 0.8141.

**Table 5  Regression relation for COD and turbidity removal.**

| $R^2$/Responses | Response 1 | Response 2 |
|---|---|---|
| $R^2$ | 0.9985 | 0.9292 |
| Adjusted | 0.9972 | 0.8629 |
| Predicted | 0.9942 | 0.7169 |
| Adequate Precision | 83.7345 | 10.7936 |

Regarding the turbidity removal, the $R^2$ and adjusted $R^2$ were 0.9292 and 0.8629, respectively. The values suggest that the model may describe 92.92% of the overall deviations in turbidity removal. Although the values were comparatively lower than those obtained for COD removal, the difference between the $R^2$ and adjusted $R^2$ values was less than 1.0, confirming the high correlation of the model between the experimental results and the predicted data. Besides, a similar lower adjusted $R^2$ trend with that of the COD removal shows that the addition of factors in studying the turbidity removal might cause overfitting.

## Optimization of COD and turbidity removal

The treatment of SS stands as a critical concern in ensuring environmental sustainability and public health. The COD (*Sedighi et al., 2018*) and turbidity (*Okoro et al., 2021*) values serve as pivotal indicators that reflect the organic pollutants and suspended particles present in water bodies. The primary purpose of this investigative study is to dissect and analyze the nuanced effects of parameters (temperature, agitation speed, fermentation duration, and sample concentration) on the removal of COD and turbidity from SS produced by palm oil mills. Understanding how alterations in these parameters influence removal efficiency is essential for optimizing the treatment process, reducing operational costs, and minimizing environmental impact. Through systematic experimentation and analysis, the intricate relationships between these parameters and the COD and turbidity removal could be unraveled.

A key factor in improving sustainability in the palm oil sector is the removal of turbidity and COD from the separator sludge. Reducing COD levels is vital to protect nearby ecosystems and prevent water supply pollution following sludge disposal. Effective COD removal also encourages the production of biogas and lowers the carbon footprint of the palm oil industry while promoting renewable energy sources. Apart from that, removing turbidity helps to convert sludge into high-value wastes for various applications, such as composting or enriching soil, which saves waste and increases resource efficiency. Reducing turbidity from sludge also safeguards local habitats, improves community relations, and preserves water resources by adhering to eco-friendly practices. Ultimately, removing COD and turbidity from palm oil mill sludge is essential to promoting sustainability and ensuring the sector complies with the standard regulations and practices good environmental stewardship while advancing resource recovery programs. Hence, the following sections discuss the effects of each parameter on the removal of polluting attributes while demonstrating and analyzing the physical and chemical changes that occurred during the treatment process.
## Effect of parameters on COD removal

The experimental and predicted values of the responses for COD and turbidity removal are presented in Fig. 3. The experimental measurements were then used to estimate the regression coefficients and develop the model's regression equations for both responses (Table 6) comprising linear and interaction effects. In the equations, the positive signs denote the components' synergistic effects, while the negative signs denote the factors' antagonistic effects on the dependent variable. Synergistic effects of experimental elements refer to specific scenarios where the combination or interaction of several variables results in a combined effect that is larger or different from the sum of their separate effects. In contrast, antagonistic effects of experimental factors occur when the combined impact of multiple variables is less than what would be expected based on the individual effects of each factor. As such, these factors work against each other, resulting in a reduction or interference in their overall effects.

The coded equation for Response 1 shows that the linear individual terms A, D, and interaction terms AB, AD, ABC, ABD, and ACD had a positive impact on COD removal. On the contrary, the linear terms B and C and interactive terms AC, BC, BD, CD, BCD, and ABCD result in a negative impact on Response 1. The findings suggest that an increase in temperature and sample concentration increases COD removal efficiency, while an increase in agitation speed and fermentation duration decreases the COD removal rate. The pollutant removal effectiveness is directly influenced by the relationship between temperature and microbial activity (*Varma et al., 2021*). Higher yields of free radicals at higher reaction temperatures enable the elimination of more pollutants (*Chen et al., 2012*; *Liu et al., 2020*). Besides, an increase in sample concentration increases nutrient availability for fungal growth. Thus, increasing fungal activity could directly improve COD removal efficiency.

Figure 3A depicts the relationship between the experimental values and the predicted values derived from the mathematical model, with most of the response residuals being relatively close to the straight line in the plots. This configuration explains an excellent match between the experimental and predicted values, which confirms the accuracy and reliability of the model. The perturbation plots in Fig. 4 show the interaction and impact of relevant factors on the experimental data. Plots of perturbations can be used to visualize the resilience of a system or model to changes in its inputs and determine specific parameters that need to be managed or changed to maximize efficiency or minimize the effect of uncertainty. The slope in Fig. 4A indicates a progressive rise in COD removal for variables A and D. This suggests that the interaction of A and D is statistically significant for COD removal. Essentially, altering factors A and D from low to high levels increases the COD removal rate.

Three-dimensional (3D) graphs called response surface plots were used to understand the relationship between two decision factors and the response variable. The impact of temperature, agitation speed, fermentation duration, and sample concentration and their interactions on COD removal is shown in Fig. 5. The response surface plots consist of the blue and white zones, which represent the maximum and minimum potential degradation, respectively. As depicted in Fig. 5, the effect of temperature and agitation speed at a fixed

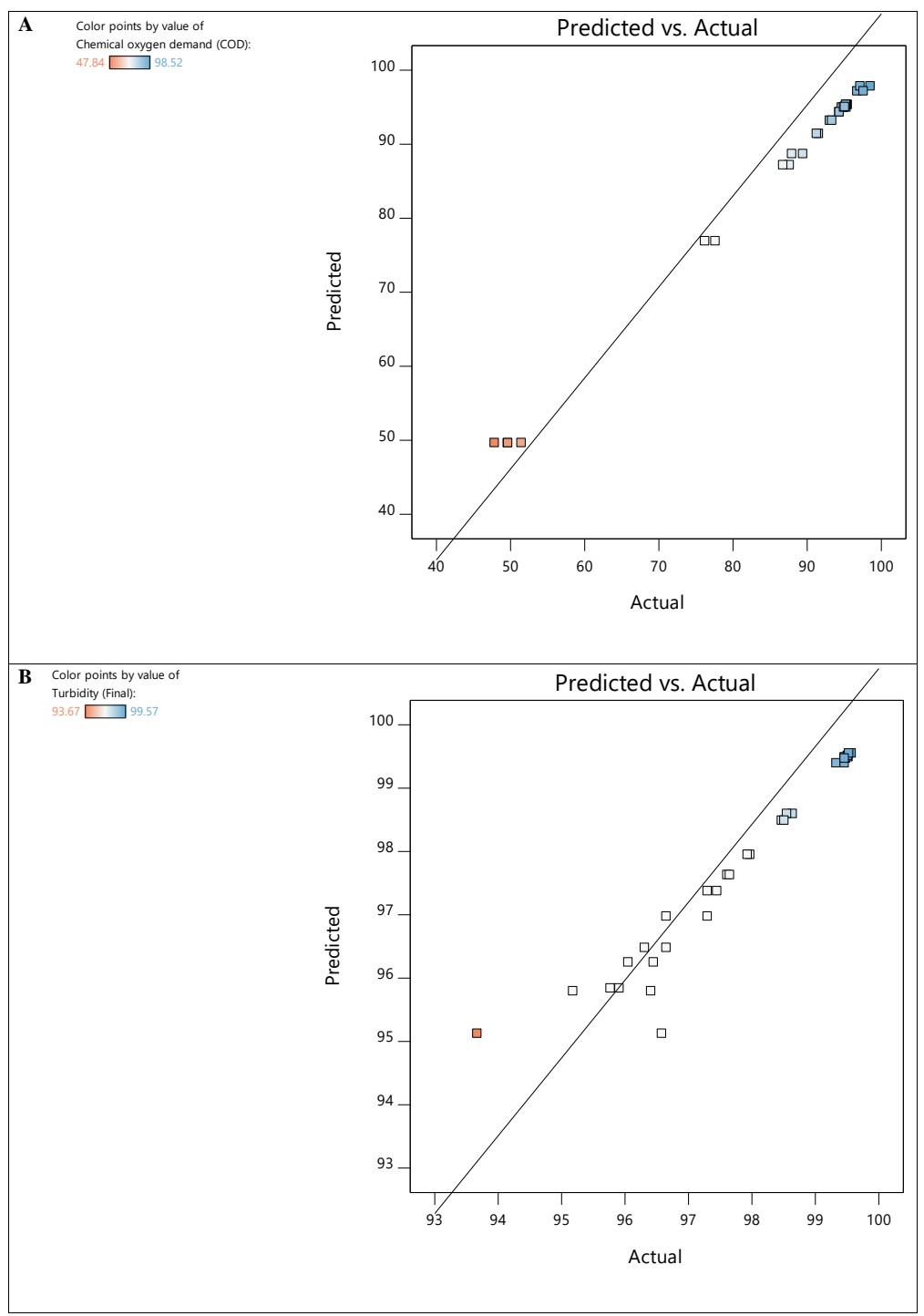

**Figure 3** **Predicted *vs* actual values plot for COD (A) and turbidity (B) removal.**

**Table 6   Regression equations in terms of coded factors.**

| Response | Final coded equation |
|---|---|
| 1 | $87.3116 + 4.81594 * A + -0.150312 * B + -0.153438 * C + 1.61344 * D + 1.09281 * AB + -0.872813 * AC + 0.362812 * AD + -3.25281 * BC + -6.17969 * BD + -5.18156 * CD + 7.29406 * ABC + 4.01094 * ABD + 6.93406 * ACD + -2.01469 * BCD + -1.29531 * ABCD$ |
| 2 | $97.745 + -0.226875 * A + -0.196875 * B + -0.12 * C + -0.434375 * D + 0.52375 * AB + -0.023125 * AC + -0.25 * AD + 0.520625 * BC + 0.1775 * BD + 0.553125 * CD + 0.3675 * ABC + -0.398125 * ABD + -0.18875 * ACD + 0.315 * BCD + -0.734375 * ABCD$ |

fermentation duration (156 h) and sample concentration (60%) signifies that the COD removal efficiency increases with increasing temperature and agitation speed. Previous studies also revealed that higher temperatures improved contaminant removal effectiveness (*Chen et al., 2012*; *Liu et al., 2020*). In fact, a review by *Varma et al. (2021)* reported that lower temperatures result in poorer COD removal. However, most of the degradation processes involving microorganisms have limitations in terms of temperature, as higher temperatures above 40 °C lead to denaturation of biological components. Exceedingly high temperatures during fermentation might also cause harmful intracellular alterations and increase the death rate of fungal cells. In contrast, slower fermentation occurs when the temperature is set too low. In a recent study, the optimal temperature for *A. niger* ranged from 34.9 °C to 37.4 °C (*Dagnas et al., 2017*; *Dagnas, Onno & Membre, 2014*) and can therefore explain the impact on the COD removal. Hence, it is best to maintain the reaction temperature of 37–40 °C for an optimized COD removal.

In addition, *Kucharczyk & Tuszyński (2018)* suggested that higher temperatures shortened the fermentation duration and promoted the synthesis of by-products, such as alcohols, esters, and acetaldehyde. Thus, treated sludge could be further studied to enhance the recovery of value-added products. Besides, performing the fermentation at 37–40 °C enables the sample to be treated at 100% concentration without the need for dilution, ensuring sustainable and energy-saving biological treatment. On the other hand, the effect of agitation speed showed a contradicting trend. The plots obtained suggest that a combination of high agitation speed and temperature results in higher COD removal. However, a lower agitation speed is required when combined with higher-concentration sludge. Based on this, agitation must be carried out at an ideal speed to maximize the interactions between the sample and the adsorption sites on the adsorbents in the mixture. Overall, optimal COD removal was achieved at high temperatures and high sample concentrations with lower agitation speed and shorter fermentation duration.

### Effect of parameters on turbidity removal

Regarding the regression equation for Response 2 in Table 6, each individual term has a negative impact on turbidity removal. Nevertheless, the interactive terms AB, BC, BD, CD, ABC, and BCD exhibit a synergistic impact on turbidity removal. The results clarify the more significant turbidity removal rate at lower temperatures, agitation speeds,
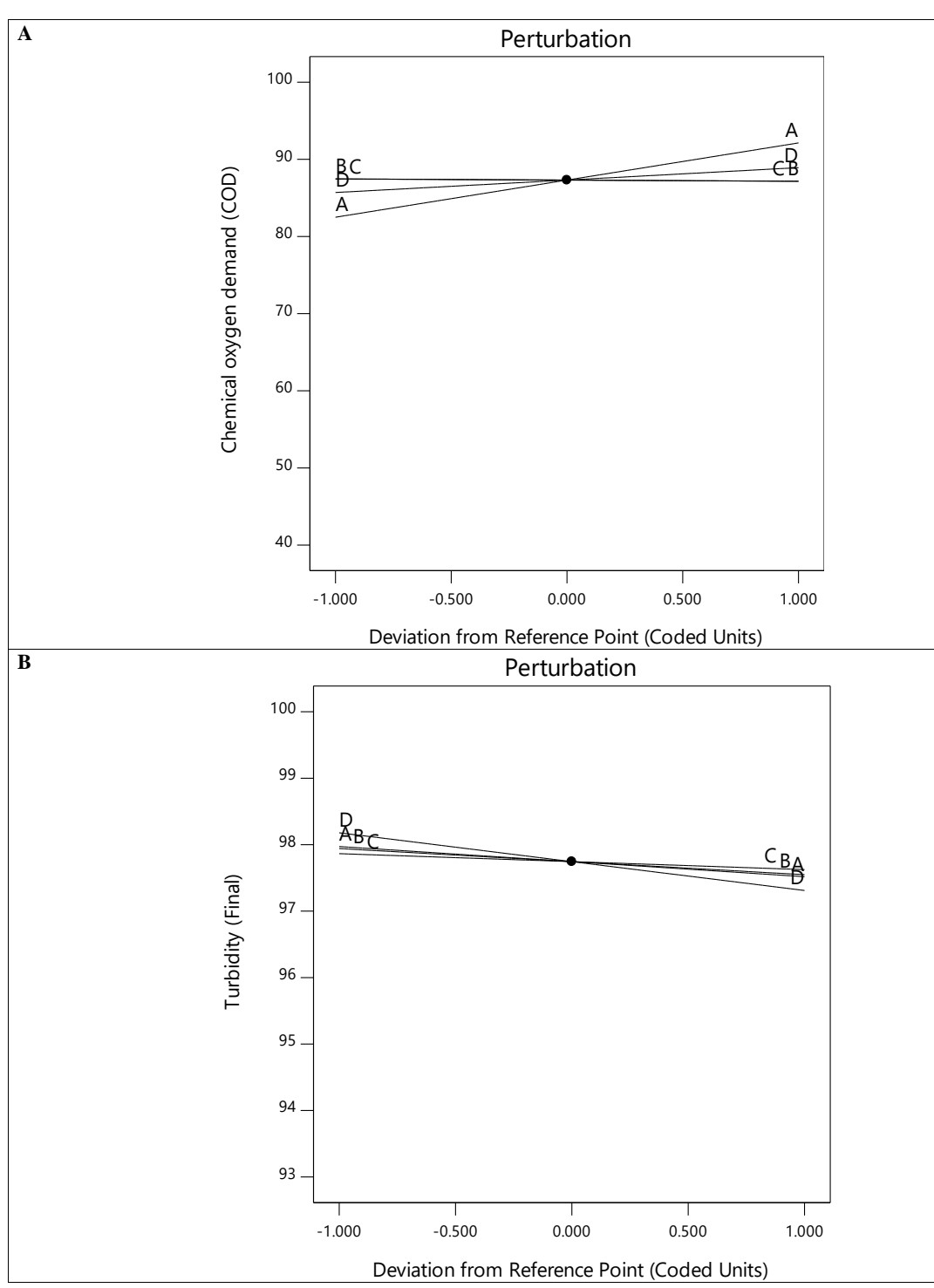

**Figure 4  Perturbation plot for COD (A) and turbidity (B) removal.**

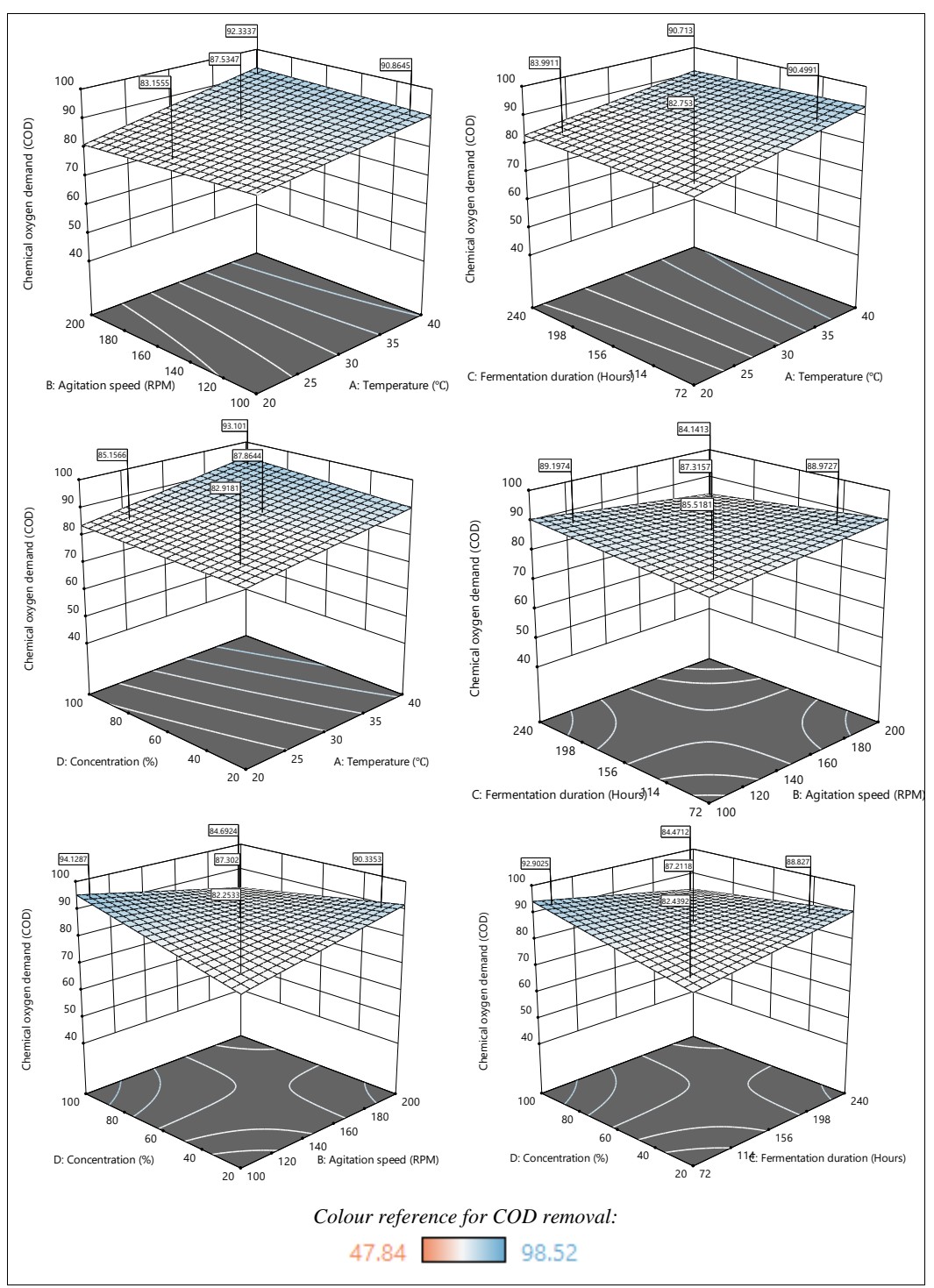

**Figure 5** 3D response surface plots of the effect of parameters on COD removal.

fermentation duration, and sample concentration. The perturbation plot in Fig. 4B shows that turbidity removal increases with the alteration of variables A, B, C, and D from low to high.

The response surface plots in Fig. 6 show the impact of parameters A, B, C, and D on turbidity removal. The turbidity removal plots show less color variation compared to the plots presented for COD removal, especially those that represent the relationship between temperature and fermentation duration. Nevertheless, the response surface plots indicate a higher turbidity removal as temperature, agitation speed, fermentation duration, and sample concentration decrease. Specifically, the response surface plots depict more significant turbidity removal at lower temperatures (20–25 °C) and agitation speeds (100–120 RPM). Although the fermentation duration had a minimal impact on turbidity removal, the highest removal was recorded during the shortest fermentation period of 72–156 h. Besides, higher turbidity removal was achieved with a lower initial sample concentration regardless of the temperature change. A similar finding was reported, where a lower temperature range (2–30 °C) increased the turbidity removal rate (*Dayarathne et al., 2022*). The turbidity removal efficiency is also temperature-dependent because the rise in temperature reduces the viscosity and speeds up the settling. Another study supported the idea of maintaining the agitation speed within the suggested range (100–120 RPM) because the flocs formed during coagulation and flocculation were easily broken at lower mixing speeds (below 80 RPM) (*Ernest et al., 2017*), decreasing the removal efficiency. Conversely, the coagulation efficiency did not increase due to the greater floc shearing at faster mixing speeds (over 160 RPM).

Numerous filamentous fungi species, including *A. niger*, have been reported to self-pelt *via* coagulative or non-coagulative methods (*Gultom, Zamalloa & Hu, 2014*). The non-coagulative process is mediated by the hyphae that grow from the spores, forming pellets. In contrast, spores mediate the coagulative process. The coagulative mechanism that took place due to the spore's dispersal and development within the initial fermentation period, known as the exponential phase, explains the higher turbidity removal within 72–114 h. Finally, the plots obtained show efficient turbidity removal at a lower initial sample concentration. *Antov et al. (2018)* also recorded an efficient turbidity removal at lower initial wastewater concentrations. However, dilution is required to lower the initial sample concentration, which in turn increases the treatment cost. Thus, it is pertinent to consider the factors and their impact when optimizing the treatment process.

## Point prediction and verification

The ideal process parameters for the most significant removal of COD and turbidity in SS have been identified using numerical optimization evaluation. As demonstrated from the experimental results, the removal efficiency of COD and turbidity falls within the range of 47.84–97.59% and 93.67–99.57%, respectively, at an initial temperature (A) of 37 °C, agitation speed (B) of 100 RPM, fermentation duration (C) of 156 h, and sample concentration (D) of 100%. These conditions were keyed in the design expert software under the selected desirability conditions between −1 and 1. It is essential to understand that choosing a desirable number between −1 and 1 is not a requirement that must always

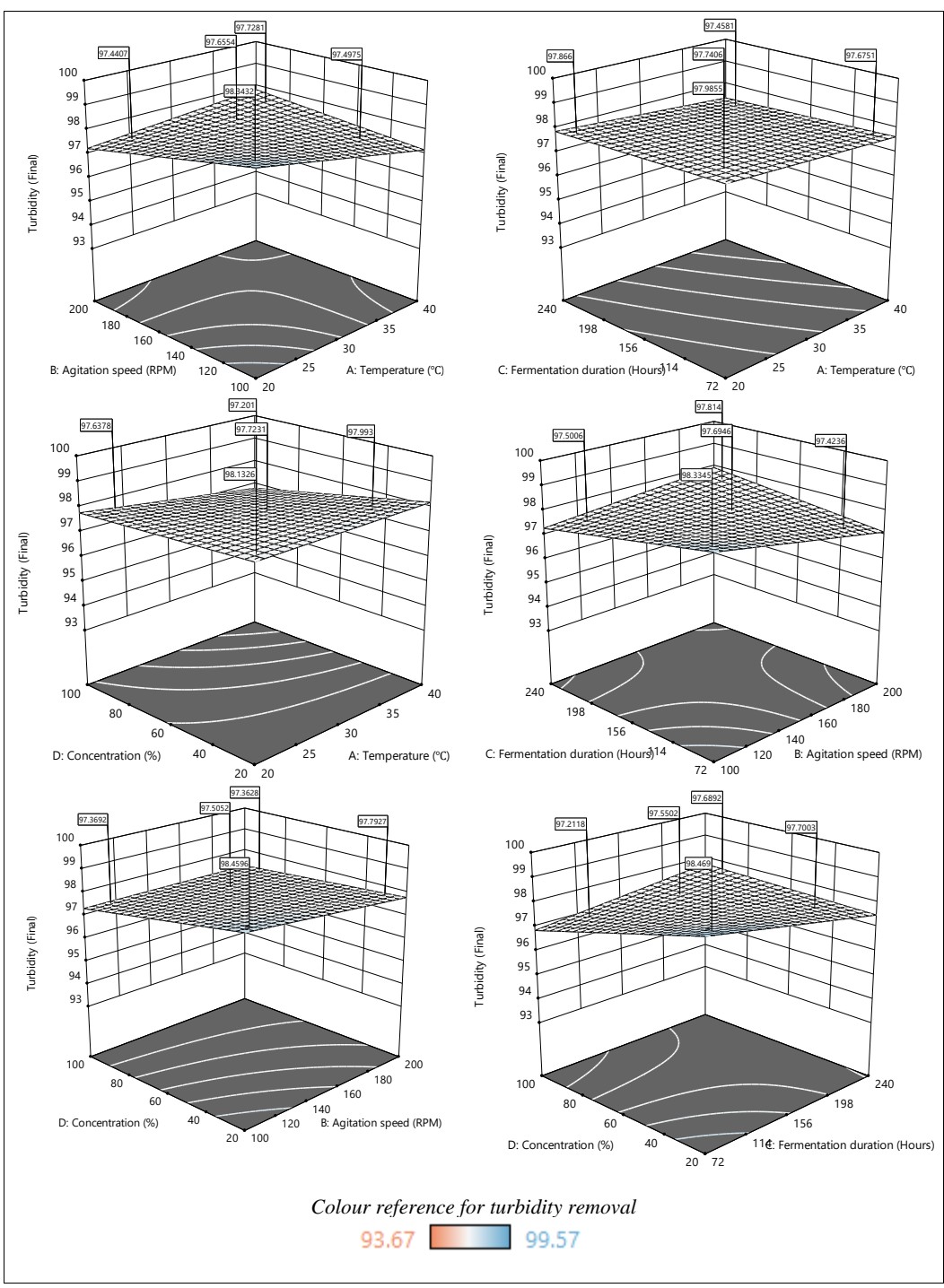

**Figure 6** **3D response surface plot of the effect of parameters on turbidity removal.**

be met but depends highly on the proximity of the lower and upper bounds to the real optimum value.

**Table 7 Verification experiments at optimum process conditions.**

| Response | Predicted | Experimental | Predicted error, % |
|---|---|---|---|
| Chemical oxygen demand (COD) | 95.3048 | 97.43 | 2.23 |
| Turbidity (Final) | 96.9083 | 95.11 | 1.86 |

**Table 8 Effects of bioremediation on the physicochemical properties of sludge.**

| Parameters | Unit | Raw sludge | Treated supernatant | Reduction |
|---|---|---|---|---|
| Ph | – | 4.61 | 4.48 | 0.13 |
| COD | mg/L | 192, 900 | 4, 940 | 97.43% |
| BOD | mg/L | 28, 500 | 8, 500 | 70.18% |
| Turbidity | FAU | 15, 680 | 766.7 | 95.11% |
| OG | mg/L | 1079.4 | 267.97 | 75.17% |
| Colour | – | $L^* = 5.60$ | $L^* = 41.03$ | $\Delta L^* = +35.43$ |
| | | $a^* = 2.11$ | $a^* = 18.53$ | $\Delta a^* = +16.42$ |
| | | $b^* = 6.75$ | $b^* = 47.94$ | $\Delta b^* = +41.19$ |
| | | | | $\Delta E^* ab = 9.65$ |
| TC | mg/L | 38640 | 20620 | 46.64% |
| TOC | mg/L | 38590 | 20560 | 46.72% |
| TIC | mg/L | 57.4 | 57.5 | −0.17% |

The results demonstrate that the biological treatment of SS was successfully optimized with enhanced bioremediation. Table 7 also presents only a slight discrepancy between the predicted values and the experimental results, which reinforces the idea that RSM in a two-level factorial design is a valuable tool for establishing stable operating conditions for the laboratory-scale bioremediation of SS using *A. niger*. On top of that, achieving less than a 5% error signifies the high precision and accuracy of this bioremediation study, enhancing its reliability and credibility. The result also supports replicability and bolsters the trustworthiness of the findings, with implications for decision-making and resource efficiency.

## Overall pollutant mitigation

Table 8 summarizes the removal efficiency of COD (97.43%), BOD (70.18%), turbidity (95.11%), OG (75.17%), TC (46.64%), and TOC (46.72%) after 156 h of fermentation period under optimized conditions. COD and BOD are the most crucial constituents used as indicators of organic pollution in water bodies. The present study's results indicate a greater COD removal compared to the maximum COD removal in POME of 83.66 ±1.9%, as reported by *Karim et al. (2021)* using a co-culture inoculum comprising *Bacillus cereus* and *Lipomyces starkeyi*, while the performance of individual cultures yielded lower removal rates of 74.35 ±1.7% and 69.01 ±2.3%, respectively. In another study, *Binma-Ae, Saek & Kayeeyu (2021)* treated POME with *Aspergillus* sp. and achieved a BOD and COD removal of 31.02% and 49.50%, respectively, which was much lower than the results obtained in the current study.

Besides, a slight decrease in the pH level caused the sample to become slightly more acidic, possibly due to the acidic metabolites produced by *A. niger.* In a previous study, *A. niger* was utilized in the fermentation of POME, which recorded the production of CitA, oxalic acid, and gluconic acid. Then, the reduction of TC and TOC could be related to the consumption of carbon sources by *A. niger*, which led to the removal of COD, BOD, OG, and color (*Neoh et al., 2013*).

The color measurements, CIE L⋆, a⋆, and b⋆ values for the raw SS were 5.60, 2.11, and 6.75, respectively (Table 8). The values for L (lightness), a (redness), and b (yellowness) in this coordinate system vary from 0 (black) to 100 (white), −100 (greenness) to +100 (redness), and −100 (blueness) to +100 (yellowness), respectively (*Mokhtar, Swailam & Embaby, 2018*). SS appeared darker where it is more red and yellow, as implied by the positive a⋆ and b⋆ values. In contrast, the post-treatment CIE L⋆, a⋆, and b⋆ values were 41.03, 18.53, and 47.94, respectively. It can be deduced that the optimized bioremediation has a significant impact on the color removal of SS. Therefore, it is crucial to perform pre-treatment of SS prior to sludge disposal, as direct disposal to the environment will trigger a detrimental impact on the ecosystem.

In summary, the highest COD and turbidity removal was achieved with less than 5% error from the expected values at 97.43% and 95.11%, respectively. Comparatively, the removal percentages achieved in this study are close to the results reported by *Soo, Bashir & Wong (2022)* using an IAAB system, which is around 99%. Additionally, the bioremediation under optimized conditions decreased the levels of other polluting parameters, such as BOD, OG, color, and carbon content. This work establishes the significant reliability of the two-level factorial design in identifying domain factors and maximizing the bioremediation of SS using *A. niger*. Despite the promising results, the experiments were conducted at a laboratory scale, which may not fully represent the complexities and challenges of implementing the proposed optimized SmF in real-world industrial settings.

## Degradation mechanism

The sequence of physical, chemical, or biological processes that cause a substance to break down or deteriorate over time is referred to as the degradation mechanism. Several factors are involved in this process, including the surrounding environment, chemical processes, and biological activity. Therefore, this study focused on examining high-resolution images of the surface morphology of SS *via* SEM to facilitate the visualization of structural and morphological modifications following the degradation processes. The degradation process also facilitates the observation of changes in functional groups. In view of this, the presence of chemical bonds and functional groups in SS was determined using FTIR spectroscopy. Chemical degradation can be indicated, for instance, by bond cleavage, the formation of new functional groups, or variations in peak intensities. The formation of new chemical species or compounds during degradation can also be identified using FTIR. Thus, understanding the breakdown products or intermediates produced during the degradation process is aided by spectral change analysis.

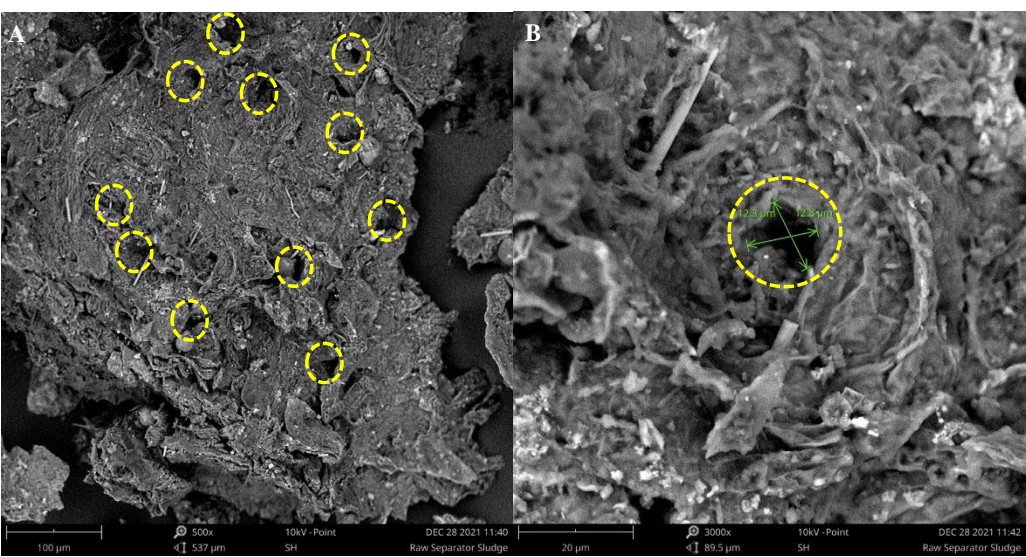

**Figure 7  SEM images of raw SS.** (A) Raw SS magnified by 500× with pores marked. (B) Raw SS magnified by 3000× with a pore marked and measured. (Diameter ≈ 12.3–12.8 μm).

## SEM analysis

Figures 7 and 8 present the SEM images of the raw and treated SS, respectively. Uniquely, both the raw and treated SS revealed the presence of fibrous needle-like structures. In addition, freeze-dried raw SS depicts a rough yet compact and non-homogenous surface with irregular pores. In contrast, more pores were developed on the surface of the biologically treated SS. The fermentation using *A. niger* also doubled the pore size. As a larger and higher number of pores increases the surface area, these characteristics render the treated sludge appropriate for biotransformation *via* pyrolysis and hydrothermal process. Thus, treated sludge could be reused as low-cost sludge-based adsorbent using carbonization, physical activation, and chemical activation. Further phytotoxicity study would assist in identifying the suitability of the treated sludge as soil conditioner and fertilizer. SS, which is rich in organic matter, could also be used to improve soil structure and water-holding capacity, making it more resilient to drought and erosion.

## FTIR spectra analysis

Figure 9 shows the FTIR analysis of the bioremediation of SS using *A. niger* in the 4,000–600 cm$^{-1}$ range and the relationship between raw and treated SS (Table 9). Functional group region within 4,000–1,500 cm$^{-1}$ in the raw SS shows two major peaks at 3,295.19 cm$^{-1}$ and 1,638.96 cm$^{-1}$. The former indicates the presence of a high concentration of alcohol, O–H (phenols). This corresponds with the natural composition of POME, which contains a high concentration of hazardous phenolic compounds (*Boontham, Habaki & Egashira, 2020*). The dark brown color of POME is caused by the oxidation of phenolic components, such as lignin and anaerobically broken-down products. This also explains the high concentration of phenol in SS, as it is one of the major components in POME. A similar peak was observed

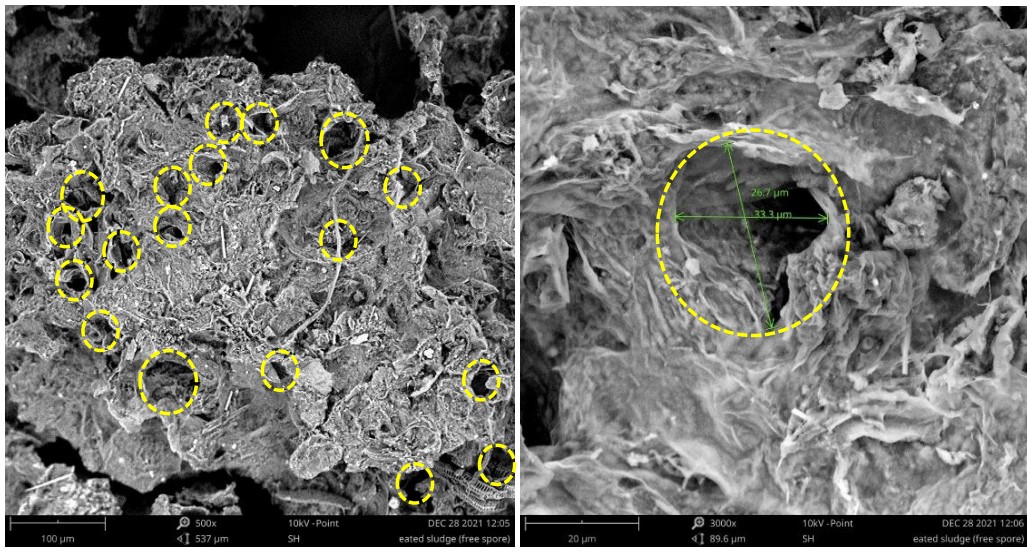

**Figure 8** **SEM images of treated SS.** (A) Treated SS magnified by 500× with pores marked. (B) Treated SS magnified by 3000× with a pore marked and measured (Diameter ≈ 26.7–33.3 μm).

**Table 9** FTIR spectra of key functional groups and their correspondence wavenumber (cm⁻¹) of raw and treated SS.

| Wavenumber (cm⁻¹) | Bond/Stretching | Class/compound | Observed wavenumber (cm⁻¹) | |
|---|---|---|---|---|
| | | | **Raw SS** | **Treated SS** |
| 3,200–3,550 | O-H | alcohol | 3,295.19 | 3,291.15 |
| 2,850–2,960 | C-H | alkane | – | 2,917.93 |
| | | | | 2,851.07 |
| 1,500–1,700 | C=O and N-H | Amide | 1,638.96 | 1,629.57 |
| 1,163–1,210 | C-O | Ester | 1,187.79 | – |
| 1,395–1,440 | O-H | Carboxylic acid | – | 1,413.00 |
| 1,000–1,400 | C-F | Fluoro | – | 1,030.67 |

in the treated SS at 3,291.15 cm⁻¹ with higher transmittance (lower absorbance), which indicates a decrease in phenolic components in the treated SS.

Meanwhile, the peak at 1,638.96 cm⁻¹ suggests few possibilities, for example, the presence of nitrile group, C=N, which has a similar conjugation effect to carbonyl group, C=O, or strong primary amines, N–H. A similar peak at 1,629.57 cm⁻¹ was detected in the treated SS, indicating the presence of exchangeable protons, which often originate from amide, amine, alcohol, and carboxylic groups. These groups' stretching serves as proof of the presence of lignin's carbohydrate, protein, and fatty acid constituents (*Alvarez-Vazquez, Jefferson & Judd, 2004*; *Lanan et al., 2021*). Other than that, the treated SS shows two distinct narrow peaks at 2,917.93 cm⁻¹ and 2,851.07 cm⁻¹, which represent alkane groups, indicating changes in the chemical compound due to the presence of *A. niger*.

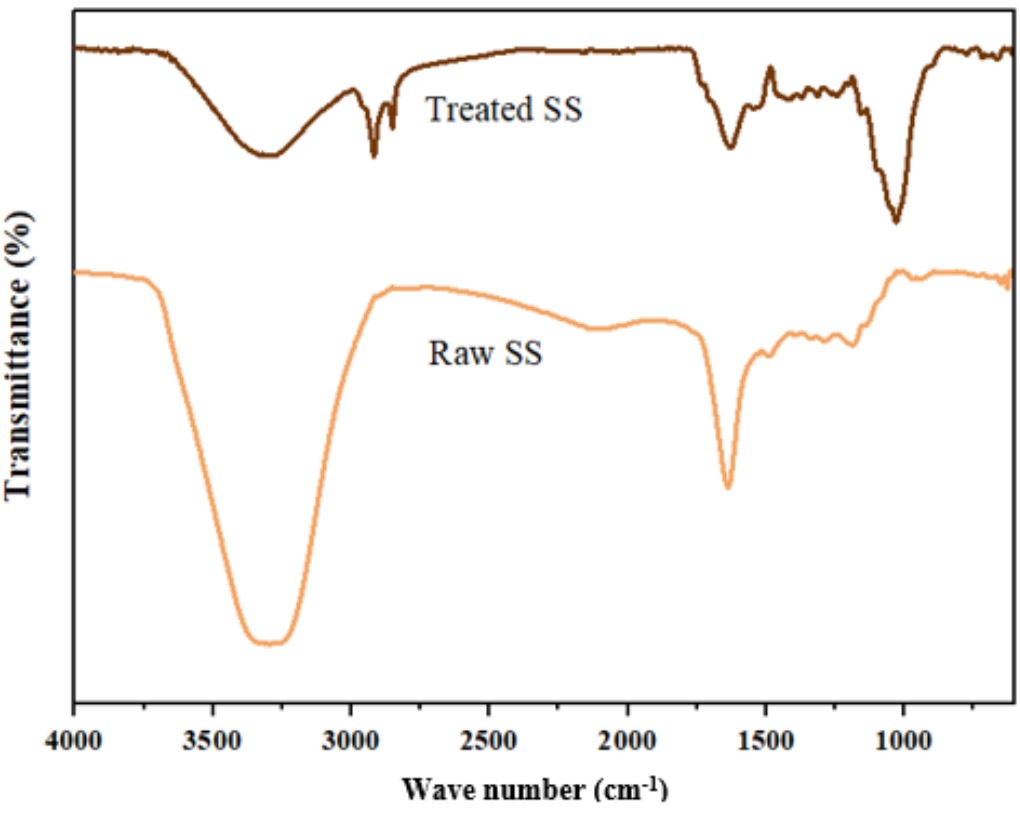

**Figure 9 FTIR spectrum for raw and treated SS.**

Analysis of fingerprint regions of the raw SS shows a single peak at 1,187.79 cm$^{-1}$, which indicates the presence of ester. Further characterization of the ester could help in developing biofuel using SS as an alternative cheap raw material. In contrast, a band at 1,413 cm$^{-1}$ in the treated SS could imply the presence of carboxylic acid. Another strong band observed at 1,030.67 cm$^{-1}$ in the treated SS indicates fluoroalkane. In short, the fermentation of SS by *A. niger* substantially altered its physical and biochemical makeup.

The biochemical composition of the treated SS could be further analyzed, and targeted components could be assessed through biochemical assays to identify the possible bioconversion activities during bioremediation. SmF of SS could also generate value-added products due to the presence of various nutrients. Commercially, *A. niger* is used to produce extracellular enzymes, such as pectinase, protease, amylase, amyloglucosidase, hemicellulose, and catalase, which are crucial components in the food industry (*Jayasekara & Ratnayake, 2019*). In addition, *A. niger* can produce lipase, one of the most significant and often utilized enzymes in industrial processes, such as those in the biofuels industry, which benefit from its capacity to hydrolyze fat.

According to *Putri et al. (2020)*, using organic waste from agro-industrial processes as a substrate offer more financial benefits, given that *A. niger* cellulases can break down cellulose in fabric fibrils. Combinations of additional enzymes, including lipase, protease,

and amylase, have also been explored for their practical application as detergent additives. Utilizing waste feedstocks instead of ''pure'' feedstocks is frequently more cost-effective in chemical processes and is typically driven by the demands among companies for waste materials with cheaper removal and treatment in place of those that would otherwise. Most significantly, this study addresses a particular waste stream of the palm oil industry that has turned into a significant sector in many nations, which now poses a major challenge from an ecological point of view. When properly applied, circular economy strategies could make a substantial contribution to the UN Sustainable Development Goals (*Waudby & Zein, 2021*).

## CONCLUSION AND RECOMMENDATIONS

This study explored the effect of fermentation parameters, such as temperature, agitation speed, fermentation duration, and sample concentration, in achieving optimal COD and turbidity removal using the two-level factorial design. Selected parameters of the SmF were expected to reduce the pollutant level in SS significantly. The results demonstrated that all variables were significant ($p < 0.05$) in achieving optimum bioremediation. This statistical tool was suitably applied to determine the domain factors affecting the bioremediation of SS generated from palm oil mills. The findings also indicate that *A. niger* is highly efficient in removing contaminants and is a promising bioprocess tool for the bioremediation of wastes with high organic content. The data obtained in this study could be used for conducting large-scale research to assess the feasibility and efficiency of bioremediation for industrial quantities of SS and investigate the long-term environmental impacts and sustainability of the bioremediation method. Besides, other approaches to convert the by-products of bioremediation into valuable materials or energy sources should be explored. Finally, research on meeting regulatory standards and permits for bioremediation processes and a comprehensive environmental assessment of using treated sludge in various applications will ensure the sustainability of the bioremediation method.

### Funding
The research was supported by the Ministry of Education (MOE) through the Fundamental Research Grant Scheme (FRGS/1/2020/STG05/UM/02/11). The funders had no role in study design, data collection and analysis, decision to publish, or preparation of the manuscript.

### Grant Disclosures
The following grant information was disclosed by the authors:
Ministry of Education (MOE) through the Fundamental Research Grant Scheme: FRGS/1/2020/STG05/UM/02/11.

### Competing Interests
The authors declare there are no competing interests.

## Author Contributions

- Paveethra Thegarathah conceived and designed the experiments, performed the experiments, analyzed the data, prepared figures and/or tables, authored or reviewed drafts of the article, and approved the final draft.
- Jegalakshimi Jewaratnam conceived and designed the experiments, authored or reviewed drafts of the article, and approved the final draft.
- Khanom Simarani conceived and designed the experiments, authored or reviewed drafts of the article, and approved the final draft.
- Amal A.M. Elgharbawy analyzed the data, authored or reviewed drafts of the article, and approved the final draft.

## Data Availability

The raw data is available in the Supplementary File.

## Supplemental Information

Supplemental information for this article can be found online at http://dx.doi.org/10.7717/peerj.17151#supplemental-information.

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
