# Peer review of "Aspergillus niger as an efficient biological agent for separator sludge remediation: two-level factorial design for optimal fermentation"

_PeerJ, doi:10.7717/peerj.17151_

## Round 0.1 · original submission · Major Revisions

Dear Authors,

Kindly address all comments as suggested by reviewers in a point by point manner.

Warm regards,
Dr. Arindam Mitra
Academic Editor
PeerJ Life & Environment

**Language Note:** The review process has identified that the English language must be improved. PeerJ can provide language editing services - please contact us at [email protected] for pricing (be sure to provide your manuscript number and title). Alternatively, you should make your own arrangements to improve the language quality and provide details in your response letter. – PeerJ Staff

Reviewer 1 ·

Basic reporting

The article provided content effectively presents a study on bioremediation of palm oil mill separator sludge using Aspergillus niger. The background concisely establishes the significance of the study by linking the palm oil industry's growth to the mounting environmental challenges posed by separator sludge. The inclusion of experimental conditions, treatment efficiency, and pollutant reduction pathways is commendable. However, the content could benefit from more context on the specific environmental impacts caused by separator sludge and how bioremediation addresses these issues.
The methods section provides a clear outline of the experimental design, enhancing the content's scientific credibility. However, the reasoning behind the chosen independent variables could be elaborated upon. The results are well-detailed, showcasing the optimal conditions for COD and turbidity removal. However, the significance of achieving less than 5% error from predicted values could be discussed in relation to the study's reliability and accuracy. While the content emphasizes the benefits of the optimized conditions, it would be valuable to elaborate on the broader implications of these reductions in other pollutants, such as how these improvements could lead to a more sustainable palm oil industry. Lastly, the conclusion effectively summarizes the study's findings, but it could be strengthened by including a call to action or future research directions to encourage further exploration of this bioremediation approach. Overall, the content provides valuable insights into the study's contributions, but it could be enhanced by addressing the aforementioned aspects.

Experimental design

The "Materials & Methods" section presents a comprehensive approach to investigating bioremediation using Aspergillus niger. The choice of a well-established A. niger strain from the University of Malaya's Microbiology Department enhances the study's reliability. The incorporation of a two-level factorial design to optimize fermentation conditions is commendable and demonstrates a structured approach. The attempt to manipulate variables such as temperature, agitation rate, fermentation time, and sample concentration showcases a systematic effort to enhance the bioremediation process.
However, the experimental design's clarity could be improved by providing explicit information about the ranges and levels of the considered factors. The equations for COD and turbidity removal lack clarity due to formatting issues, affecting comprehension. A more detailed explanation of the purpose behind selected conditions like temperature and agitation speed would strengthen the study's scientific foundation. Additionally, the analysis section could benefit from including the interpretation of results from surface morphology and chemical functional group assessments. Considering the integration of Response Surface Methodology (RSM) might enhance the experiment's efficiency and overall robustness in optimizing the bioremediation process.
The text discusses the use of Half-Normal plots, Pareto Charts for Standardized Effect, and Analysis of Variance (ANOVA) in evaluating the robustness of a model. The application of Half-Normal plots to assess the influence of factors on COD and turbidity removal is highlighted. Factors that fall below the regression line are deemed insignificant, while those above are considered influential. The text suggests that interaction terms have favorable effects on degradation. While the analysis approach is sound, there is a need to clarify the rationale behind the chosen variables and their effects.
The use of ANOVA is mentioned to identify significant main and interaction effects impacting degradation. Although the procedure is clear, there is room to improve the presentation of results. The p-values obtained from ANOVA are crucial in determining the statistical significance of factors. The influence of each factor is discussed for both COD and turbidity removal, indicating that certain factors are statistically significant while others are not. The application of Fit Statistics, such as R-squared (R2) and adjusted R-squared, to evaluate the model's predictive capability is noteworthy. The high R2 values indicate strong correlations between experimental results and model predictions. However, there is an opportunity to provide a clearer explanation of the significance and implications of R2 and adjusted R2 values in the context of the study.
Overall, the text effectively explains the application of these statistical techniques in evaluating the model's reliability. However, further clarification on the rationale behind variable selection and a more detailed interpretation of certain results would enhance the reader's understanding.
The section in question delves into the comprehensive investigation of how various parameters influence the removal of COD and turbidity. A commendable aspect is the inclusion of both experimental and predicted response values in Figure 3, which facilitates a visual comparison of outcomes. Additionally, the derivation and elucidation of regression equations integrating linear and interaction effects showcase a proficient grasp of statistical modeling techniques.
Nonetheless, there are key aspects that warrant further refinement: Structurally, the section could be enhanced by greater organizational clarity. An introductory segment explicitly outlining the purpose of exploring parameter effects on COD and turbidity removal would offer essential context to readers. Further segmentation of the discourse, addressing individual parameters in distinct sections, could bolster overall coherence. The interpretation of findings could be substantially enriched with a more contextual perspective. Expanding on the rationale underpinning observed effects and their alignment, or lack thereof, with established theories or previous research findings would elevate the discourse's credibility and scientific rigor. Furthermore, technical terminology, such as "synergistic effects" and "antagonistic effects," might present challenges to some readers. Providing succinct definitions for such terms would ensure accessibility to a broader audience. It is suggested that the incorporation of pertinent citations would reinforce arguments. Linking current findings to existing scholarly research, elucidating commonalities or disparities, would infuse the discourse with depth and authority. Lastly, a more thorough interpretation of visual aids, such as Figures 4 and 6, would be advantageous. Delving into observed trends, significant patterns, and their alignment with the experimental design would heighten the discourse's insightful character. In summary, the section provides valuable insights into the intricate relationship between parameters and the removal of COD and turbidity. By augmenting structural coherence, contextualization, terminology elucidation, scholarly referencing, and graph interpretation, the discussion would attain a more unified and enlightening essence.

Validity of the findings

The "Results and Analysis" sections provide an extensive exploration of experimental outcomes, which is commendable for its thoroughness. The presentation of both experimental and predicted values for COD and turbidity removal showcases the meticulous nature of the study. The use of statistical tools like half-normal plots, Pareto charts, and ANOVA to identify optimal conditions is a strong analytical approach that enhances the study's robustness. Furthermore, the identification of significant factors influencing the responses demonstrates the authors' dedication to methodological precision.
Percent Improvement (%Improv):
Calculate the percent improvement by finding the difference between the experimental result and the baseline result, then dividing it by the baseline result and multiplying by 100.
Percent Improvement (%Improv) = ((Experimental Result - Baseline Result) / Baseline Result) * 100
Sensitivity Index (SI):
Calculate the sensitivity index to measure the influence of a parameter on the response. Find the change in the response divided by the change in the parameter, then multiply by the parameter value divided by the response value.
Sensitivity Index (SI) = (Change in Response / Change in Parameter) * (Parameter Value / Response Value)
Pearson Correlation Coefficient (r):
Validate optimized conditions using real-world data by calculating the Pearson Correlation Coefficient between predicted and actual results. Sum the product of the differences between each data point and its respective mean for both datasets, then divide by the product of the standard deviations of the datasets.
Pearson Correlation Coefficient (r) = Σ((X - X̄)(Y - Ȳ)) / √(Σ(X - X̄)² * Σ(Y - Ȳ)²)
Confidence Interval (CI):
Apply uncertainty analysis by calculating the confidence interval around your results. Calculate the confidence interval by adding and subtracting the product of the critical value and the standard error from the mean.
Confidence Interval (CI) = Mean ± (Critical Value * Standard Error)
Integrating these formulas into your analysis will provide a more comprehensive understanding of your study's outcomes and their implications.
However, while the analysis demonstrates a comprehensive understanding of the data, there is room for improvement in contextualizing the findings within existing research. Including comparisons with established literature would strengthen the study's significance by showcasing its contribution to the field. Additionally, although the study's design is well-executed, a more critical examination of potential limitations, such as the generalizability of results or confounding factors, would bolster the study's reliability.
The text provided exhibits a generally clear and organized structure, discussing various methods and techniques for evaluating and optimizing experimental conditions. The explanations of the methods are clear and well-defined, providing readers with a solid understanding of the concepts involved. Additionally, the formulas are accurately represented in text format, aiding comprehension.

Additional comments

However, there are a few instances where the text could benefit from enhanced clarity and grammar refinement. For instance, some sentences appear lengthy and might benefit from being broken down into shorter, more concise phrases. Additionally, there are a few grammatical issues, such as missing articles ("the" or "a/an") or minor punctuation adjustments that could enhance overall readability. Employing transition words or phrases could help establish smoother connections between concepts, enhancing the flow of the text.
To ensure optimal clarity and coherence, it's recommended to carefully review the text for sentence structure and grammatical accuracy. Furthermore, considering the use of transition words or phrases can contribute to a more seamless progression of ideas.

Reviewer 2 ·

Basic reporting

The manuscript is generally well written and clear. However, to ensure clarity and accessibility for a broad audience, acronyms should be spelled out in full upon first mention.

These include: CPO (Crude Palm Oil), SS (Separator Sludge), HCW (Hydro-cyclone Water), BOD (Biochemical Oxygen Demand), COD (Chemical Oxygen Demand), TSS (Total Suspended Solids), TN (Total Nitrogen), AN (Ammonia Nitrogen), and OG (Oil and Grease).

Additionally, there are minor sentence corrections and typos that need to be addressed.

1) Line 41 of the abstract, the word "Different" is capitalized in the middle of a sentence. It should be in lowercase, as "different".
2) Line 64: "POME are being produced" should be "POME is being produced".

3) Line 116: "characterization of the residue obtained will provide the basics of metabolites present as a proof of revalorization while degradation takes place" should be "characterization of the obtained residue will provide basic information about the metabolites present as evidence of revalorization during degradation".

4) Line 118: "bioremediation and revalorization of SS from palm oil mill for a sustainable sludge handling in the future because the ultimate objective of bioremediation is, at the very least, for the sludge to be converted into biomass or nontoxic metabolic products" should be "bioremediation and revalorization of SS from palm oil mills for sustainable sludge handling in the future, as the ultimate goal of bioremediation is to convert sludge into biomass or non-toxic metabolic products, at the very least".

5) Line 126: "it was kept in sterile containers and refrigerated at 4°C" should be "it was stored in sterile containers and refrigerated at 4°C".

6) Line 132: "the fungus strain was revived" should be "the fungal strain was revived".

These are just a few examples, and a thorough proofreading of the document is recommended to ensure all errors are corrected.

Experimental design

Your methodology section is super detailed, but some readers might not be familiar with the statistical methods you used. It might be helpful to explain these methods in simpler terms, or provide references to resources where readers can learn more about these methods. For example, you could break down what a two-level factorial design is and why it's useful for your study in a way that's easy for anyone to understand.

Validity of the findings

Here are some additional suggestions to improve the overall paper quality and impact.

1) Background Info: In your introduction, it might be helpful to go into more detail about how palm oil production and POME affect the environment. You could talk about the specific pollutants in POME and how they mess with soil and water quality. You could also explain how these pollutants can harm local ecosystems and people's health. This would help readers understand why your study is so important.

2) Next steps: In the discussion section, you could dive into the potential uses for your findings. For example, you could chat about how the treated SS could be used as a soil conditioner, a source of bioenergy, or a raw material for other industrial processes. You could also brainstorm potential markets or industries that could benefit from these applications.

3) Limitations and Future Research: Towards the end of your discussion section, it could be useful to reflect on any limitations of your study. This could include things like aspects of your methodology, the scope of your research, or any challenges you faced during the study. You could also suggest areas for future research, like testing how scalable your method is, or exploring other potential uses for treated SS.

4) Economic Feasibility: While your study focuses on the technical feasibility of using Aspergillus niger for bioremediation of SS, it could also be interesting to discuss the economic feasibility. You could talk about the potential costs associated with this method and compare it with current waste treatment methods. This would give a more complete picture of how viable your proposed solution is.

Reviewer 3 ·

Basic reporting

Review for the article by Thegarathah et al “Aspergillus niger as an efficient biological agent for separator sludge remediation: Two-level factorial design for fermentation process optimization”

This manuscript reports efficient bioremediation of palm oil mill effluent (POME) by using Aspergillus niger as bioremediation agent. Furthermore, they have studied the effect of various experimental conditions and treatment efficiencies on bioremediation. The manuscript uses a lot of abbreviations without giving full forms. Furthermore, some interacting factors responsible for bioremediation are poorly defined. This makes manuscript hard to understand.

My Concerns

Line 41 Change Different to different.
Line 49 COD give its full form and provide full form at all the places like Line 58.
CPO, SC, SS.
Line 75 Please remove the double hyphen
Line 216 What are these parameters A, B, C, D, and other parameters?
Figure 1 legend Provide more details in the legend. What does the green zigzag depict? what do the blue and red squares depict? Figure 3 What do the color-coded squares depict in A and B
Figure 5 what green, yellow, and red colors depict in this surface plot. Provide a key to the color scheme. same for the surface plot in Figure 6.
Figure 9 The y-axis should be wave number, not waven umber

Experimental design

no comment

Validity of the findings

no comment

Additional comments

no comment

---

## Round 0.2 · accepted · Accept

The reviewers' comments have been addressed. The manuscript is now ready for publication.

Reviewer 2 ·

Basic reporting

No comments

Experimental design

No comments

Validity of the findings

No comments

Additional comments

Dear Authors,

I've had an opportunity to review your revised manuscript and I must say, I'm quite impressed with the thoroughness of your revisions. Given the substantial improvements you've made, and considering the significant contribution this research brings to our field, I am in full support of this manuscript's publication.

Thank you for taking my feedback into account and for your diligence in making the necessary revisions. I eagerly anticipate seeing your valuable work published and am confident it will have a positive impact in our scientific community. Best of luck

Reviewer 3 ·

Basic reporting

I appreciate the authors for their prompt and thorough revisions addressing my earlier concerns. The modifications have significantly strengthened the manuscript, and I am now satisfied with the completeness and clarity of the content.

Experimental design

'no comment'

Validity of the findings

'no comment'

Additional comments

'no comment'